# Do Food and Meal Organization Systems in Polish Primary Schools Reflect Students’ Preferences and Healthy and Sustainable Dietary Guidelines? The Results of Qualitative Research for the Junior-Edu-Żywienie (JEŻ) Project

**DOI:** 10.3390/foods13010061

**Published:** 2023-12-22

**Authors:** Ewa Czarniecka-Skubina, Jadwiga Hamulka, Marta Jeruszka-Bielak, Krystyna Gutkowska

**Affiliations:** 1Department of Food Gastronomy and Food Hygiene, Institute of Human Nutrition Sciences, Warsaw University of Life Sciences (SGGW-WULS), 166 Nowoursynowska Street, 02-787 Warsaw, Poland; 2Department of Human Nutrition, Institute of Human Nutrition Sciences, Warsaw University of Life Sciences (SGGW-WULS), 166 Nowoursynowska Street, 02-787 Warsaw, Poland; jadwiga_hamulka@sggw.edu.pl (J.H.); marta_jeruszka-bielak@sggw.edu.pl (M.J.-B.); 3Department of Food Market and Consumer Research, Institute of Human Nutrition Sciences, Warsaw University of Life Sciences (SGGW-WULS), 166 Nowoursynowska Street, 02-787 Warsaw, Poland; krystyna_gutkowska@sggw.edu.pl

**Keywords:** primary schools, school pupils’ feeding, meal quality, school shops, vending machines, eating behavior, food preferences, individual in-depth interviews (IDIs)

## Abstract

The school environment, together with that of the family, shapes students’ eating behaviors, and is an important element of formal and informal nutrition education. The informal and practical dimensions can be realized through the food and meals offered/sold in school canteens, shops, and vending machines. As children and adolescents spend 6–10 h/day in schools and consume at least two meals there, the school food environment is important from a public health perspective. The aim of this study is to assess food and meal organization in primary schools in Poland. The study was conducted using individual in-depth interviews (IDIs) involving 24 school headmasters and 24 representatives of people employed in school canteens or catering companies involved in meal organization in schools. The nutritional food on offer at schools is important for shaping students’ food preferences and choices, consequently influencing the development, functioning, and health of this subpopulation. The school environment can be an ideal place for hands-on nutrition education. In addition to the knowledge provided, there must be a consistent message concerning the provision of nutrition information to students between the teachings of parents, teachers, and, indirectly, the food and meals available at school. Our qualitative study is confirmed by the results of quantitative research to better understand the organization of nutrition and the problems and needs of primary schools in this area.

## 1. Introduction

In accordance with its statutory obligation, the task of schools is to teach and educate children and adolescents in various areas, shaping their attitudes toward life by providing knowledge and skills, which also relate to food and nutrition. Students present a “readiness to learn” and consider school an educational institution, a source of knowledge, and a place that promotes a healthy lifestyle, including healthy eating habits [1,2,3]. A school’s food and nutrition environment may have significant impacts on students’ perceptions of healthy eating. This must be consistent with the messages provided by parents and teachers and with the contents of the nutritional programs implemented by schools to avoid cognitive dissonance. Moreover, in some instances, students who receive school meals need financial support. Many countries have a legal requirement to provide certain groups of students with free (subsidized), nutritious school lunches [4,5,6]. Also, in Poland, primary schools are obliged to provide students with one hot meal a day and give them the opportunity to consume it during their stay at school [7], as well as to provide a subsidized meal for students requiring financial help based on their household income. Moreover, the assortment of foods and meals sold in school shops, buffets, and vending machines, as well as those served in school (and kindergarten) canteens, were officially regulated for the first time in 2015 [8] and then updated in 2016 [9].

Children and adolescents usually spend 6 to 10 h in an educational institution for approximately 180 days a year. Education time equals, on average, 1620 days spent at primary school; therefore, children and adolescents should have the opportunity to eat a nutritious meal at school as well as buy a second breakfast. Breaks taken between meals that are too long and low-quality meals may have an adverse effect on students’ functioning. Therefore, the food offered at school should be of a high quality, which will have a positive impact on students’ health and psychophysical development as well as school achievements. During the developmental period, the requirement for all macro- and micronutrients increases. It should also be emphasized that this subpopulation is particularly sensitive to the negative effects of receiving inadequate nutrition, e.g., deficiencies in vitamins and minerals or the excessive intake of free sugars and saturated fatty acids, as well as total energy. Any nutritional deficiencies can cause disturbances in proper growth and development processes as well as in maintaining a proper psychophysical condition [10]. School meals provide, on average, 30–40% of individuals’ daily energy requirements and make an important contribution to all nutrients’ intake [11,12]. Thus, it is extremely important that schools offer appropriately balanced meals prepared using high-quality products that are consistent with the recommendations and assumptions of the planetary diet, guaranteeing the protection of both health and the planet, through, e.g., reductions in carbon and water footprints [13], as well as a reduction in food waste.

Primary school students usually consume 1–2 meals (second breakfast and lunch) at school. When a balanced meal is not provided and consumed, the likelihood of reaching for snacks, such as sweets and fast food, e.g., chips, with high amounts of added sugars, saturated fats, and salt, increases, which may contribute to the development of diet-related diseases, including obesity, type 2 diabetes, cardiovascular diseases, and some cancers [14].

Comprehensive school nutrition programs, including the nutrition standards of foods, meals, and drinks served at the school [15,16,17], can have a positive long-term impact on the nutritional status and health of children and adolescents, such as improving their BMIs, reducing risk factors for metabolic diseases [18,19,20,21], as well as decreasing micronutrient deficiencies [22]. Therefore, school meals should be an integrated part of the educational process implemented in school environments and may play an important role in promoting and supporting adequate nutrition among students [10,16].

Previous research has indicated the need to select high-quality products and serve balanced meals in school canteens which consider the nutritional requirements of children and adolescents. However, due to the variety of meal organization practices at school, as well as limited financial means, the economic aspects are dominant, which means that product selection and acquisition are often determined by the price rather than quality. Since children and adolescents spend a considerable amount of time in school, an important question arises—does the school food environment influence their eating patterns and can it be a source of nutrition education? The aim of this study is to identify food and meal organization in primary schools, including the availability in school shops, vending machines, typical school canteens, and food delivered by catering operators, based on the opinions of school employees and catering company representatives responsible for the foods and meals available at school obtained during qualitative research.

## 2. Materials and Methods

### 2.1. Study Design and Participants

Qualitative research was carried out in 24 primary schools chosen from 2218 schools that had signed up for the Junior-Edu-Żywienie (JEŻ) project. Schools were selected from various regions in Poland, due to the varying number of places’ inhabitants as well as students attending the school. Each school was coded with consecutive numbers. Schools from the following places participated in the research: Warsaw and its surroundings (8 schools, no. 1–8), Poznan (2 schools, no. 9, 10), Lublin (no. 11), Białystok (no. 12), Kielce (no. 13), Ostrowiec Świętokrzyski (no. 14a,b,c,d, the same catering operator), Toruń (no. 15), Tarnobrzeg (no. 16), Nowy Sącz (no. 17a,b, the same catering operator), Brańszczyk (no. 18), Muszyna (no. 19), and Ustka (no. 20), representing cities and towns, and Poręba (no. 21), Trzcianka (no. 22), Tylicz (no. 23), and Zielonki Parcele (no. 24), representing villages.

The inclusion criteria included the consent of the headmaster to conduct qualitative research in their school; the willingness and consent of employees working in a school canteen or a catering company delivering meals to a given school; and being a person responsible for meal organization in a school, e.g., an attendant, cook, or school headmaster. There were no exclusion criteria for participating in research.

IDIs were conducted with 48 participants, such as school headmasters (*n* = 24), canteen workers (attendants, cooks), and representatives of catering companies (*n* = 24). The participants were open minded, easygoing in interpersonal communication, and eager to share their professional experience and opinions on food and meal organization in schools. The interviews took place at the school. The participants were mainly women (90%), which resulted from the employment structure in Polish schools, especially staff responsible for both meal organization and management of schools.

### 2.2. Qualitative Research

The in-depth interview (IDI) technique was applied to conduct the qualitative research. This is an open-ended, discovery-oriented method to obtain detailed information from a stakeholder about a topic. Its goal is to explore a respondent’s point of view, experiences, feelings, and perspectives in depth. Participants were encouraged to speak freely on the topics included in the interview scenario [23,24,25], which had the characteristics of an issues/topics sequence in a direct conversation. IDIs were carried out by the authors of this publication according to a developed list of issues (so-called check list), which was previously verified in a pilot study and then improved according to the submitted comments.

The choice of this method allowed us to obtain in-depth explanatory information on the food and meal organization in schools. The questions in the scenario were designed in accordance with the concept of a 4-Stage Funnel Approach [26], i.e., the recommendation to lead the interview from general to more detailed issues. The scope and rules of the interview were explained to the participants before the start. The following questions were then discussed: types of meal organization, operation of school shops and/or vending machines, number of people eating meals, the type of meals preferred by students, assessment of students’ and school staff’s satisfaction with the meals offered, scale of food waste at school, conceptions of an ideal food system at school, meal organization during the COVID-19 pandemic, identification of problems in meal organization, as well as the prospects for changes and needs in this area. Each interview lasted approximately 60 min.

The interviews were audio-recorded with the consent of the participants. The statements were also transcribed, which facilitated the interpretation of the results. The transcripts and notes of the original statements were analyzed. Due to the anonymity of the study, only the name of the town was used in the analysis.

### 2.3. Ethical Approval

This study was approved by the Ethics Committee of the Institute of Human Nutrition Sciences of the Warsaw University of Life Sciences (No. 18/2022) and was conducted in compliance with the Declaration of Helsinki. All respondents gave their written informed consent to participate in the research. Data obtained during the interviews were confidential and restricted only to the researchers.

## 3. Results

### 3.1. Meal Organization in Primary Schools in Poland

The schools participating in this research had their own canteens or canteens run by an agent or a catering company. In a few schools, meals were delivered by catering companies. Half of the schools also had school shops and/or vending machines (Table 1). School shops and vending machines were less common in rural schools.

Meals in school canteens were mainly consumed by students and, to a lesser extent, by teachers and other school employees (Table 1). It was noticed that teachers from large cities took advantage of meals offered at school more often, i.e., due to the longer distances from work to home and lower prices than in canteens/restaurants outside school. All schools which had such a need implemented programs supporting children living in food insecure/low-income households, and recently also children from Ukraine. This type of support is subsidized by the government, municipalities, or municipal social welfare centers.

In all schools, a single lunch (soup or second course, or the entire lunch set) could be purchased after reporting this fact in advance. Some students ate meals only on certain weekdays, which proves the great flexibility of the canteens.

Younger students (in grades 1–3) consumed meals under the supervision of a teacher sometimes during the lesson time, while older students ate meals independently during longer breaks dedicated to a hot meal. In all schools, there was always a teacher on duty who kept order in the canteen. It was observed that fewer people used school meals in December and June due to the large number of days off. Mealtimes were often flexible, as students had one to three lunch breaks during which they could eat meals. In turn, in the case of catering, lunch was served during one or two breaks, but fewer children had school meals.

The number of staff employed in a canteen varied and ranged from three to eight people, which varied with the size of the school and the number of meals served as well as the financial capabilities of the municipalities (Table 1). They were employed in the positions of attendant/provider, cook, and kitchen assistant. Most of them were women, culinary school graduates, or people who had previously gained culinary experience and appropriate training. They worked 8 h daily, mainly from 6 a.m. to 2 p.m. or from 7 a.m. to 3 p.m.

The percentage of children and teaching staff consuming meals at school depended mostly on the quality of the meals, including their organoleptic properties (palatability) and price. In schools where the meals were tasty and cheap (school codes: 1, 6, 12, 16, 18, 23; Table 1), the percentage of consumers was high, between 65.6 and 90.1% of students (Table 1 and Table 2). However, in schools where prices were higher (school codes: 9; 14a,b,c,d; 17a,b; 20, 22; shown in Table 1), students often did not eat school meals (percentage of users varied between 5.7 and 25.7%). When such situations resulted from students’ financial problems, teachers informed the school headmaster or pedagogue about the need to provide those students with a free lunch. Schools are obliged to implement programs supporting children from the poorest households. It should also be emphasized that in many schools, parents were willing to make additional payments to Parents’ Councils in order to subsidize students who needed support.

Although schools do not monitor the reasons for not eating school meals, the respondents indicated that students often gave the reasons as follows:-‘distasteful food’.-‘being on special diet’.-‘possibility of eating a meal at home after returning school’.

However, the main reason for teachers not eating lunch at school was the inconsistency of their class schedules with lunch breaks and the price.

The electronic lunch payment system introduced in one of the schools was a great convenience for parents and allowed them to order lunch one day in advance, even if the child had not eaten in the school canteen before. Moreover, the system of canteen cards introduced in this school made serving meals easier as the students activated their card in a special reader before the meal, informing the staff that the meal was paid for.

Examples of school/canteen/catering employees’ statements:
‘A canteen is needed, in younger grades almost all students eat lunch’.(Brańszczyk)
‘In a large school with catering operator, organizing meals is very difficult, some products cannot be offered because it will be at the expense of the dish quality’.(Lublin)

### 3.2. Meals Offered in Primary Schools

Lunch was the meal most frequently offered in school canteens (Table 2). Schools with kindergarten students offered full board meals consisting of breakfast, lunch, and an afternoon snack for the youngest children.

Schools offered both single-course and full meals at very diverse prices, starting from PLN 4 and ending with PLN 17, mainly in large cities, where the costs of raw foods and transport are higher. Higher prices for school meals were observed in schools with less funding from local government authorities.

Only a few schools offered meals for children with special nutritional needs (allergies, special/elimination diets), which was a result of the unprofitability of preparing special dishes for individual people, particularly in the case of school canteens. An agreement to warm up meals brought from home in a school microwave was mentioned as a possible solution. The whole lunch, only soup, or only the main course could also be purchased, which was appreciated by some parents. Many schools offered only one-course lunches due to the necessity of offering a cheap meal for students.

Meals were usually served during two or three, and in a few schools during four, 20 min breaks, depending on the number of places available in the lunchroom. Lunch was served starting at 11 a.m., and in one school even at 10.30 a.m. Catering companies delivered lunches in thermoports or in single-portion insulated containers between 11 a.m. and 2 p.m., depending on the school (Table 2). Then, meals were served immediately by catering or school workers (mostly in rural areas) in the lunchroom or, when such places were unavailable, in designated rooms, e.g., common rooms.

Example of a school/canteen/catering employee’s statement:
‘The more people take the meal, the greater the financial possibilities of preparing varied meals’.(Poręba)


### 3.3. Supplies for Canteens and Catering Companies

The number of catering service providers depended on the size of the city/town and the number of people eating meals, as well as on the supplier’s capabilities, e.g., specializing in specific staple foods, semi-finished products, or having a wider range of products, e.g., suppliers specializing only in bread, meat, or vegetable and fruit supply.

The choice was usually made using a trial-and-error method until a good supplier was found and the cooperation could last for years. Most schools used regular, trusted local suppliers, local bakeries, sometimes farms, and chose Polish products (vegetables, eggs from farms). Suppliers were selected as a part of public procurement procedures. Unfortunately, the price, not the quality, mainly determined the choice of supplier, which did not guarantee adequate attractiveness of meals to consumers, which in turn often resulted in a small number of people eating the meals, and thus an unsatisfactory profit for the entity responsible for meal organization in the school.

Deliveries of dry products, such as pasta, flour, etc., were carried out according to the agreed deadlines, and fresh products and bread were delivered every day. Typically, meat was delivered three times a week and frozen foods were delivered once a week.

Many schools used deliveries from wholesalers, and sometimes they took advantage of promotions in supermarkets to reduce the cost of meals, especially in smaller towns where there was no choice of food wholesalers. Most of the schools (with one exception) were public ones, so according to the law, all purchases had to be made through public procurement procedures.

Examples of school/canteen/catering employees’ statements:
‘The selection of foods is the most important, searching for good contractors and ensuring that there are no highly processed products. It is also important to use seasonal fruits and vegetables’.(Warsaw)
‘We use local suppliers, farms, and domestic products, looking for the best and cheap raw resources, semi-finished and final products’.(Poznań)

### 3.4. How Are the School Meals Planned?

Schools participating in this research planned menus for 10 days. Nutritional recommendations and dietary guidelines were considered when planning menus. Only one school used products according to a seasonal calendar. Staff took advantage of free trainings, choosing offers with the shortest duration. Sending someone for a two-day or longer training would make every-day work difficult in the canteen due to the small number of employees. The staff usually used professional materials/books for planning menus, modifying recipes to suit their own needs. None of the schools used menu planning software. Schools occasionally used the services of external dietitians, but they did not employ them as regular staff dietitians.

Recommendations for school menus were implemented through the following:-Reducing salt addition by adding fresh or dry herbs.-Reducing sugar addition or optionally sweetening with honey.-Including seasonal fruit and vegetables (e.g., pumpkin).-Following special diets (e.g., lactose-free or gluten-free for students with allergies).-Limiting frying to once a week or giving up the frying process by using combi-steam ovens, boiling, and stewing.-Using rapeseed oil for frying.-Ordering eggs from nearby farms in smaller towns/villages.

Ordering diets, including elimination ones, was only possible in one canteen, where lunches were prepared by a catering company. Only two schools offered a choice out of two meal options, so if a child did not like one option, they could choose another dish.

Catering companies usually employed one person as a dietitian who planed menus in accordance with nutritional recommendations. The school only approved the menus.

In other schools, an attendant, especially an enthusiast who increased their knowledge and skills during trainings, tried to implement nutritional recommendations within the proposed meal prices. Only one sport school employed a dietitian. In one of the schools, the attendant and the school nurse planned the menus. In some schools, people with dietetic or nutrition education were also employed as attendants.

Employees often used the information shared on the Facebook group of attendants. They exchanged their experiences and recommended tested recipes there. In some cities, such as Warsaw, the municipality organized regular trainings for canteen staff. These included trainings on hygiene, the implementation of HACCP principles, and ensuring adequate dietary quality of dishes. Trainings were also offered by sanitary inspection facilities.

Nevertheless, it is difficult to implement all nutritional recommendations within the meal rate proposed by the school. In order to meet the financial criteria, schools decided on serving either one-pot meals or a main course and a soup with meat added interchangeably.

Examples of school/canteen/catering employees’ statements:
‘In addition to following dietary guidelines, we try to tailor meals to children’s preferences; parents pay and expect their children to eat meals’.(Warsaw)
‘We try to make the children eat this lunch’.(Toruń)

### 3.5. Dishes Liked and Disliked by Students

Based on the statements obtained during interviews, it can be concluded that children preferred some dishes and ate others reluctantly, which resulted from habits acquired at home. Many students did not know certain flavors at all and only learned them in school canteens (Table 3).

According to staffs’ statements, it can be assumed that the most favorite soups included vegetable soup, cucumber soup, red borscht, cauliflower soup, pea soup, bean soup, cabbage soup, krupnik soup, żurek soup, and cream-type vegetable soups (Table 3). The most popular were tomato soup and broth with noodles.

Main courses often consisted of a meat, a starch, and vegetables. The most liked meats included pork chops (although these were rarely offered in school canteens) and pork tenderloin, chicken in various forms (thighs, breasts, in sauce, e.g., chicken in sweet and sour sauce), and minced meat (cutlets, meatballs). The meat was grounded by canteen staff to decrease the costs and increase the sensory quality. However, beef cutlets were disliked and were offered quite rarely due to the high prices of beef and the children’s difficulties in chewing. Children’s favorite dishes also included one-pot dishes such as spaghetti Bolognese.

A fish dish was offered once a week, although it was disliked by students. Baked salmon or salmon soup were the exceptions. In some schools, children did not come to the canteen on Fridays when fish was usually served, or even parents did not buy meals for those days. In order to encourage students to eat fish, varying fish dishes were prepared like fish cutlets, meatballs, dumplings in sauce, fish soups, baked fish with vegetables, and fish in dough. Preparing fish dishes in a more attractive way helped to increase fish consumption.

Potatoes, various types of groats, rice, pasta, and potato dumplings were usually served as starches. Fries were prepared occasionally and only to please the students.

Vegetables in the form of salads were not liked by the students, except for fresh cucumber salad (Polish ‘mizeria’) and carrot salad with apple or pineapple. Lettuce, sauerkraut, fresh cabbage (coleslaw-type salad), iceberg lettuce with peppers, and celery were also quite popular. From the statements given, it can also be concluded that the students were not forced to eat salads to avoid discouraging them from eating disliked vegetables and causing negative emotions. Cooked vegetables were more preferred by students, particularly carrots with peas, beans, beets, broccoli, and cauliflower. In order to increase the consumption of vegetables, they were ‘smuggled’ in soups or meat and vegetable dishes, e.g., dishes with minced meat mixed with vegetables (meatballs with carrots, zucchini) and casseroles with spinach or broccoli. In some schools, two options were offered, a salad and a cooked vegetable, so that the child could choose what they liked. In one school, vegetables were served on platters, thus introducing a kind of competition between students, and encouraging them to try unknown dishes, including salads.

Typically, one meatless day was planned during a week and then sweet farinaceous dishes were prepared like pancakes (especially pancakes with strawberries), dumplings with cheese, potato dumplings, sweet pasta (e.g., with strawberries and cheese), or sweet rice (with apples).

For dessert, easily divisible fruit was usually offered, like apples, pears, kiwis, and bananas, which was sometimes delivered as part of a campaign by local suppliers who provided fruit for free.

From time to time, special events were organized in the canteens and sweet rolls were offered for dessert, donuts were offered on Fat Thursday, chocolates/Advent calendars were offered on St. Nicholas Day, and ice cream or other sweets were offered on Children’s Day. Regular suppliers sometimes delivered sweets for free or at a lower price, and although these were sporadic situations, they were remembered by both students and canteen staff. On special occasions, pizza, hamburgers, and sweet desserts were offered, and at festivals grilled sausages were offered. Even though this is not consistent with dietary guidelines, students appreciated such opportunities and became attached to the canteen and the recommendations given by the cafeteria staff.

Culinary techniques like traditional boiling, stewing, or baking were used most frequently. Frying was used a maximum of once a week just to please the students, especially boys, with their favorite pork chop dishes.

In line with observations made during IDIs, canteen workers ‘put a lot of heart’ into trying to prepare more attractive meals and meet the expectations of young consumers. The staff used various incentives to encourage students to eat healthy foods, especially vegetables and fish, through posters and other educational materials displayed in canteens, interesting names (‘Shrek’s soup’ for spinach soup, ‘The Undersea World’ for fish dish), as well as nutrition contests. They also tried to conduct educational activities through conversations with students, teaching them what should be eaten and why, or encouraging them to try at least a little of the dishes, e.g., salads, that children were reluctant to eat. Those examples show a lot of empathy for children, especially those with eating problems like picky eating or neophobia.

Due to the need to serve meals quickly, a vegetable/salad buffet was not recognized as a good solution. Older children ate practically everything, while younger children were more ‘picky eaters’, specifically if they were not taught to eat various foods at home or had not attended kindergarten before.

It can be concluded that children generally liked the canteen workers who were their nutritional authority and treated them like aunties. At the end of the school year, many students gave them greeting cards and flowers, which proved that their work was appreciated.

Examples of school/canteen/catering employees’ statements:
‘Younger children eat better. Older students, if they don’t like some dishes, simply don’t come to eat. Some parents try to motivate their children to eat school lunch, while others believe that if a child does not like a dish, he/she should not eat it’.(Toruń)
‘Children bring eating habits from homes where little salad is consumed. Parents often take students for fast food after school, and there is a contradiction in what they expect from canteens and what they offer at home’.(Lublin)
‘On Fridays, fewer children eat meals because there is fish for lunch’.(Warsaw)

### 3.6. Assessment of School Meals and Dishes and the Current State of School Feeding Organization

The school headmaster and/or the Parents’ Council usually monitored the canteen menus and the assortment in school shops. There was not even one incident that they intervened to change the canteen menu.

In all schools, parents could follow the menu on the school website, the notice board at the entrance to the school, or in the school e-Diary. As indicated, especially by school headmasters, parents were generally satisfied with school meals, although they raised certain demands. They opted for more vegetarian, gluten-free, and lactose-free dishes, which resulted mainly from new trends and not necessarily from their children’s health requirements.

Satisfaction with catering and good prices meant that pupils ate meals eagerly in such schools. On the contrary, when the price was too high, especially in rural areas, school meals were mainly used by Ukrainian children and those who had their meals subsidized. It was noticed that school canteens operated by agents worked best, as opposed to using external catering; in particular, when using companies who deliver meals to institutions of different types, worse meal quality and particularly organoleptic complaints were stated.

Generally, there were no cases of clear dissatisfaction with the catering offer. In smaller towns, it was often a long-term cooperation, and a consensus was developed over the years regarding the type of meals and delivery. Parents’ low interest in purchasing school meals or a lack of kitchen facilities were the main reasons for using catering services. The most common complaints were related to the organoleptic quality of meals delivered in thermoports, which was worse and required faster serving.

In general, canteens operating on school premises enjoyed a good reputation, as students were often happy to eat a tasty school lunch again on Monday after a weekend spent at home. If other situations were observed by staff or reported by students, the people responsible for meal preparation usually tried to solve the problems. According to canteen workers’ notions, for some children the breaks were too short to eat a whole meal, and then the staff enabled them to eat their soup and main course separately during two consecutive breaks.

It can be concluded from opinions expressed during the IDIs that the school meals were rather liked by the students as well as the school staff, because quite many school employees, including teachers, consumed meals in the canteen, especially in large cities. The staff learned about children’s satisfaction directly when serving meals. If children did not like a dish, they left it on the plate or asked not to be served foods they disliked. On the contrary, when children liked some dishes very much, they asked for a second helping. School meals, apart from being healthy, must be tasty and look attractive.

Information about meal satisfaction was usually obtained personally from direct consumers. Parents mostly posted their opinions on social media or reported them to teachers or headmasters, who forwarded this information to the canteen workers.

In one school (in Warsaw), a survey was developed to collect opinions among children, parents, and employees. The survey included twelve questions regarding the quality and tastiness of meals; preferences for soups and main courses, including salads; and propositions of the dishes they would like to have on the school menu. The answers were then analyzed and incorporated into the menus. In one school, parents could come to the school canteen and try the dishes and drinks, which representatives of the Parents’ Council took advantage of.

In most schools, the assessment of menus was carried out once by a dietitian, and this limitation resulted from the necessary savings it provided. Menus and health safety were assessed in all schools once a year by a state sanitary inspection.

Moreover, the percentage of consumers, specifically students, eating school meals can be a measurable indicator of the quality of the meals.

Examples of school/canteen/catering employees’ statements:
‘Prepare it in a such way that you can eat it yourself’.(Białystok)
‘Sometimes children complain about school meals because the dishes are varied, and at home they only eat what the family members like’.(Lublin)
‘Most parents and children are satisfied with school meals, what is evidenced by the constantly growing number of consumers in the canteen’.(Poręba)
‘The students are content. We listen to them, we observe them. If a dish doesn’t suit them, we try to change it’.(Warsaw)
‘Parents are happy and ask for recipes so they can prepare our dishes at home’.(Zielonki Parcele)
‘Children sometimes do not know flavors other than those of processed foods which results in difficulties with adaptation to dishes in canteens’.(Białystok)

### 3.7. The Scale of Food Waste at School

The primary reason for food waste in school canteens was the reluctance of students to eat certain dishes. Salads as well as fish dishes were mainly wasted. In most schools, children were encouraged to try smaller portions of foods they did not like and come back for more if they liked the dish. Additionally, the majority of schools offered extra helpings in order to reduce the scale of food waste.

In large cities, colloidal mills were installed in the sinks, so after grinding, some of the waste ended up in the sewage system. There was not much food waste, because the number of prepared portions was adjusted to the orders, and when some food remained, they were given to children in need. Many schools introduced a system of canceling school meals due to the absence of students every morning, which enabled them to regulate the number of meal portions needed that day. Unserved lunches, especially soups, could also be eaten by hungry children even if they did not have a lunch purchased. So, only the leftovers on consumers’ plates were collected as food waste, which was not a very large amount since relatively small portions were served.

Food waste was sorted after lunch and was collected either as municipal waste or as bio-waste by companies specialized for this purpose. In rural areas, food waste was taken to feed animals, and it was estimated to amount to approximately 10 Liters per day. Only two schools (no. 6, 18) participating in the study had their own composter, where food waste was turned into fertilizer.

Food waste was collected in a similar way when meals were provided by catering operators. It was also noticed that in the case of external catering, the school was often left with the problem of food waste disposal.

In one school, uneaten and unspent lunches were handed over to the community center. Parents could also pick up a packaged lunch if their child was sick or had to leave school before lunch time.

None of the schools had a special program or activities aimed at reducing the scale of food waste. However, reporting children’s absences in the morning, which resulted in the preparation of fewer portions, could be considered such a solution. The most food was wasted after the pandemic because, during feeding at home, children were accustomed to eating pizza and other meals ordered from catering companies.

Examples of school/canteen/catering employees’ statements:
‘After the COVID-19 pandemic, we threw away a lot of food. Children got used to pizza and other processed foods ordered by their parents and it was difficult for them to switch to canteen meals’.(Toruń)
‘Many parents do not report, or report too late, that their child will not eat a meal that day and these meals remain and have to be used’.(Warsaw)

### 3.8. Providing Meals in Schools during the COVID-19 Pandemic

There were no problems during the COVID-19 pandemic, as most children had remote learning. After returning to schools, the functioning of canteens was reorganized and the rules were tightened, e.g., as follows:-More breaks were organized so fewer children stayed in the canteen at one time;-For younger children, meals were delivered to classrooms and served during lessons so the children could eat in peace;-Tables and chairs were disinfected after each group, the dining room was aired, and breaks between individual groups were extended;-Dishes were served to tables instead of common serving at the window, also known as the Polish system;-The number of consumers at tables was reduced;-Children were given cutlery instead of taking it by themselves;-Staff wore gloves, masks, and face shields when serving meals;-Plexiglass curtains were installed in canteens.

Many problems occurred in canteens then. There were no rooms to organize meals for students at the same time, and meals had to be served at certain times, which was related to the children’s stay at school and, in many cases, transport organization from school. Serving times increased as children were served meals at the table and the number of canteen employees did not increase during that time. However, this was the only way to limit contact among children and reduce the risk of infection.

In schools during the pandemic, the size of groups was reduced, tables were disinfected, and groups came under the supervision of a teacher. The obligatory quarantine was perceived as the main financial problem for companies, as the information about the need for quarantine was provided practically the day before. In most schools with a catering operator, meals were not delivered during the pandemic.

The headmaster or class teachers provided information about the quarantine or isolation of classes to the canteen and to parents via the school e-Diary the day before. Therefore, meals were not prepared for those classes at that time, which limited food waste.

During the pandemic, some schools offered takeaway lunches with an extra payment for biodegradable packaging.

### 3.9. Identification of Problems and Risk Factors in Meal Organization in Primary Schools

Particularly important problems faced by primary school headmasters in the field of meal organization were as follows: employees (staff); local conditions; kitchen and lunchroom equipment; and financial resources allocated for feeding purposes. From the perspective of the consumers as well as food and meal suppliers, the proposed prices and costs incurred and reimbursed were most often indicated.

At the end of the in-depth individual interview, participants were asked about their opinions regarding optimal forms of school meal organization and ideal (model) solutions.

Canteen employees. 

In Poland, the number of people employed in school canteens is the responsibility of the local government and varies in different regions of the country. The basic problem of the operation of ‘own’ canteens in schools was the small number of full-time positions and, at the same time, the limited time to serve meals due to the short stay of children at school and transport organization in smaller towns and villages. Moreover, any staff absences, e.g., due to illness, disorganized work and made it difficult to function. In such situations, there was a rush and nervousness that was unfavorable for the atmosphere of eating meals. It must be admitted, however, that despite this, employees organized their work well, and in such cases they usually offered one meal, a soup or a one-pot dish, or a main course, alternately. The small number of staff also prevented the use of stationary training outside the place of residence. In this case, the situation was solved with efficient cooking equipment. The most frequently reported problems included staff shortages; this problem was not reported only in schools employing 6–7 people. Such shortcomings were the result of low earnings and, therefore, a lack of interest in working in a canteen or catering. One canteen used the State Fund for the Rehabilitation of Disabled Persons, which additionally subsidized disabled people employed in the canteen.

The staff also pointed out the excess of bureaucracy and the lack of kitchen help, which is necessary especially during the lunch breaks to maintain order in the canteen when serving meals.

Local conditions. 

In many schools, local problems with providing appropriate space for students to eat meals were identified. This was specifically experienced after changes in the Polish education system and the accumulation of age groups in primary schools. Moreover, a lack of rooms such as a scullery, storage rooms, or social rooms was also reported. Kitchens were usually cramped, and a few had adequate space; nevertheless, the staff could cooperate due to the small number of workers, their huge experience, and efficient cooking equipment. There was also too little usable space for catering companies due to the growing demand for their services. Sometimes there was no space to serve meals in schools where catering operators delivered meals.

Kitchen and lunch room equipment.

In some schools, the kitchens were very small, and they needed reconstruction to conveniently separate them into a washing room, pre-treatment space, and space for social purposes. The statements obtained in the in-depth individual interviews indicated that many schools faced problems with kitchen equipment. Other reported problems included the need to conduct renovations, complaints about outdated cooking equipment, and even the lack of small equipment such as pots, knives, cutting boards, and dishes (plates, cutlery, cups), which was a result of the increasing demand for school meals. Many schools noted the need for peelers, shredders, robots, egg disinfection devices, combi-steam ovens, or dishwashers with short washing times, which would make the work much easier. No needs for kitchen equipment were reported only in one school as the kitchen had been completely renovated and was equipped with modern accessories by the local municipality.

Deficiencies in the equipment in the consumer part were also mentioned. Only few schools had child-friendly wall colors and nice tables appropriate for students’ ages (lower tables for younger children). Moreover, in many schools a lack of shelves/lockers for students’ backpacks was reported. Such problems were not indicated at all only in three schools because the canteens had been renovated.

Financial aspects of school meals—costs of prepared meals and their reimbursement and prices for consumers.

It should be emphasized that despite the relatively low prices of school meals, unsatisfactory funding from local governments was underlined. On the other hand, good cooperation with suppliers was appreciated, as they sometimes provided free products, and thus the schools managed financially and the meals were tasty and met the dietary guidelines for children and adolescents.

Support for schools included, for example, ‘Program for schools’, which provides fruits and vegetables as well as dairy for free and is financed by the Agency for Restructuring and Modernization of Agriculture (ARMA).

Generally, if there were problems with suppliers, the school terminated cooperation with them and looked for new ones. Catering companies, due to a lack of funding, bore all the costs; hence, the prices in their case were significantly higher.

Optimal forms of school meal organization and proposed model solutions.

School canteens were indicated as a better solution for school meal organization from the perspective of both the school and the students, especially in the case of the latter due to the lower price and higher quality. Some schools employed people from Ukraine to provide temporary assistance. None of the schools planned any changes in school meal organization. A quite interesting solution was the possibility to purchase single meals, rather than the entire monthly package, or to purchase part of a lunch, e.g., a soup or main course. In addition, parents expressed their willingness to make additional payment for children in need because they appreciated the benefits of eating meals with their peers. Usually, the school counselor or class teachers knew who needed financial support and who was hungry and reported such needs.

Examples of school/canteen/catering employees’ statements:
‘There is a constant shortage of workers, illnesses occur, and in the case of such a small number of employees, it is difficult to organize the work’.(Warsaw)
‘It would be ideally with no cost constraints, but with more staff and better equipment’.(Warsaw)

### 3.10. School Shops and Vending Machines

The introduction of legal regulations [8,9] regarding the food products permitted (and, indirectly, prohibited) in school shops, vending machines and buffets, resulted in a relative improvement in the offer. Special emphasis was put on lowering the amounts of added sugars, salt, and fat in the food available in schools. The assortment was monitored by the school headmaster or a designated person, as well as by the Parents’ Council.

Despite the above-mentioned regulations, products that did not meet the determined criteria were still present in school shops, including salty snacks (e.g., crackers, popcorn, salty sticks) and sweets (e.g., chocolate bars, chocolate waffles, buns, donuts, lollipops). At the same time, apple crisps, cereal bars, matzo biscuits, rice wafers, nuts, seasonal fruits, and fruit salads appeared.

Most school shops also offered sandwiches, which was beneficial if a child did not bring a second breakfast from home. The price of a sandwich was similar to the price of a lunch in many places, ranging from PLN 3.5 to PLN 5. The sandwiches were made to order to ensure their freshness. Hot dishes like toasts with cheese or hot dogs could also be purchased.

The available drinks included tap and mineral water; plain, and flavored milk and yogurts; fruit and vegetable juices; as well as hot drinks like mint and fruit teas and ersatz coffee served with honey and milk. Unfortunately, sugar-sweetened beverages like flavored waters and sparkling orangeade were also available. In all schools, all students had access to water, e.g., from springs or in the canteens. Students also carried water in filtered bottles.

In some schools, vending machines were present. Overall, the assortment of vending machines improved, but they still offered salty snacks, chips, chocolate bars, jellies, and sugar-sweetened beverages. Children could also buy water, juices, dried vegetables, and fruit from vending machines.

## 4. Discussion

Changes in lifestyle and parents’ work style are the main reasons for the decline in the number of shared family meals, which in turn increases the importance of the food offered to children and adolescents in schools. It is important to have appropriate policies regulating the foods and meals available on school premises (in school shops, vending machines, canteens), which should be in line with the dietary guidelines and limit the availability of unhealthy products [27,28]. However, as the obtained results indicate, such policies require further adjustments, regular inspections, and the development of nutrition education programs to increase the knowledge and awareness of all school actors. School is an important institution for formal and informal nutrition education [10], involving varied stakeholders [29,30,31,32] and providing an optimal food and nutrition environment.

Food and meal organization in primary schools.

Currently, in Poland, there are several forms of meal organization in schools, including ‘own’ canteens, canteens operated by an agent or a catering company, as well as meals delivered to schools by catering companies. In these terms, it seems challenging for schools to provide all willing children and adolescents with high-quality meals as well as for local governments to support schools in achieving this purpose. In the case of ‘own’ canteens, municipal or city governments finance the costs of maintaining the canteen (costs of the premises, energy, water, etc.) and staff salaries. For this reason, meals offered in such canteens are cheaper because parents only pay for the so-called ‘boiler input’, i.e., raw foods, products, and semi-finished products used for dish preparation. In other cases, meals are fully financed by parents or institutions subsidizing meals for ‘the poorest’.

This study showed that the meals offered in school canteens considered students’ preferences and not always the dietary guidelines. A certain ambivalence was noticed in the nature of the actions taken, because, for example, to eliminate food waste and thus to reduce the amounts of plate leftovers, meals were prepared to meet the students’ preferences, which often did not comply with the recommendations. This type of practice was noticed by other researchers. Although Iranian guidelines for a ‘healthy’ school canteen prohibit offering, e.g., fries, cookies, crackers, ice cream, fried foods, sweet drinks, hamburgers, pizza, and confectionery, these products were available in schools [30,32].

In various countries, despite the obligatory regulations regarding the implementation of quality standards in school canteens, insufficient monitoring of the meals served in canteens was detected, and the lack of meeting the standards resulted mainly from the high prices of healthy foods [33].

School shop and vending machine assortments.

Many studies indicate that students buy snacks and sweetened beverages in school shops and vending machines, and the products most frequently purchased include sweets, chocolates and cookies, chips, and lollipops [12,34,35,36,37,38,39]. The availability of such foods in school shops proves that the guidelines regarding the food offered in schools are not being followed [40]. The WHO has recognized school tuck shops as an effective setting for improving children’s micronutrient intake [41].

It was shown that schools had to overcome many barriers to sell healthier foods on school premises. Such foods were less popular among students (perceived as less palatable and more expensive); thus, they were more difficult to sell and often expired earlier. Therefore, school shop owners may be reluctant to offer healthier foods as this contributes to food waste and their loss of income [36,42,43,44].

In Spain, the assortment of vending machines included only 10.5% recommended products (water, milk, and dairy products) and 80.5% food and drinks with high energy, fat, or sugar content and low vitamin and mineral content [45,46]. The availability of unhealthy drinks and foods in school vending machines contributes to unfavorable eating behaviors among teenagers [47,48].

A single intervention in a primary school, such as creating a store with healthy food products, will not improve the nutritional environment in schools, but it will be a stage in the implementation of this idea. At the same time, multiple approaches are recommended and the nutritional awareness of all school actors, namely headmasters, teachers, canteen staff, students, and their parents, should be increased [49].

According to the Theory of Planned Behavior, an individual’s healthy eating behavior is predicted by personal behavioral beliefs. The consumption of sugar-sweetened beverages and snacks has been shown to be associated with attitude, perceived behavioral control, and intentions to consume beverages and additional meals [50]. Students’ healthy choices are influenced by a combination of nutrition education classes with factors such as the greater availability of low-calorie food products in school vending machines, food labeling, and lower prices of healthy food in schools [51].

In schools where nutritional policies aiming to improve the nutritional quality and availability of healthy foods were implemented, the nutritional quality of food consumed by students improved and the sales of recommended snacks and drinks increased [52,53,54]. However, this was also combined with price reductions and increased promotional marking [52]. After implementing the healthy beverages policy, 79% of drinks offered in school vending machines was found to be compliant with the policy [55].

Food waste.

Our study did not show large-scale food waste due to the preventive actions taken, such as appropriate planning, introducing the possibility of informing the school about skipping a meal on a given day, donating food to community centers or hungry students, preparing dishes liked by students, and enabling second helpings for students.

The effectiveness of these types of activities is confirmed by the results of other studies. In Great Britain, operational, situational, and behavioral causes of food waste in schools were identified [56]. It was found that students did not eat the food provided by catering companies for reasons like crowded eating places, short eating times, unattractive and unfamiliar dishes, and spending lunch breaks playing instead of eating [56,57]. Other authors also pointed out the anxiety around eating, peer pressure, and influence, and the need for social interaction [57].

Our own research showed that students were reluctant to eat salads or fish, leaving them on their plates or simply throwing them away. According to Wilkie et al. [58], 20–50% of food waste consisted of vegetables and fruits. This is mainly related to the fact that children do not know such foods and are reluctant to try them. The authors also emphasized that when students brought food from home, less of it was wasted because mainly liked products were taken [58].

For this reason, it is recommended to conduct consumption analyses and satisfaction surveys with meals in the canteen, as this may help to optimize school feeding.

Improving the quality of school feeding.

Based on the data obtained from in-depth individual interviews, it can be concluded that the schools used offers from local suppliers, who sometimes delivered food products free of charge, relatively often, which allowed them not only to save money, but also to offer students fresh products, e.g., seasonal vegetables and fruits.

The quality of meals in school canteens depends not only on the quality of the food products but also on the skills and engagement of the staff. Therefore, it is crucial to improve the competences of the staff responsible for planning and preparing meals, primarily in the areas of applying quality criteria in procurement, cooking attractive and balanced meals using good-quality products, using seasonal and local products, planning meals and using surplus food to prevent food waste, applying appropriate culinary techniques and using modern cooking equipment, and changing the methods of meal serving (e.g., one-dish lunches, fruit and vegetable buffets).

It is also recommended to apply ready-made menus developed by experts that include not only the healthy criteria but also the sustainability issues. Planned menus should also be evaluated with quality indexes, e.g., SMI-LE (The School Meal Index-Lunch Evaluation), whose usage is quite simple, quick, and does not require professional equipment [59].

All undertaken activities should be guided by the idea that the meals and food products offered at schools are part of nutrition education, indicating the practical application of theoretical solutions.

### Limitation

The conducted study has limitations that are inherent to qualitative research, which, due to its small sample size, are not representative but help to better understand quantitative outcomes. However, when combined with the results of quantitative research, which is the case for the Junior-Edu-Żywienie (JEŻ) project, a deeper and more comprehensive explanation of the quantitative research findings was provided.

## 5. Conclusions

Qualitative research allowed for understanding the specificity of school food and meal organization from the perspective of the suppliers (canteen employees, representatives of agents and catering companies) and the recipients of food services (school headmasters), considering the perceived problems and proposed solutions in this area.

It has been shown that the school environment can be an excellent place for nutrition education in practice. In addition to the knowledge provided, there must be a consistent message concerning the provision of nutrition information to students between the teachings of parents, teachers, and, indirectly, the food and meals available at school. The availability and attractiveness of healthy offers increase the opportunities for students to consume foods and meals that meet the dietary guidelines. It is not an easy task due to children and adolescents’ organoleptic preferences and the higher costs of healthy foods. In schools that received funding from local governments, more attractive meals were prepared, and the number of students and teachers eating school lunches was higher. Thus, it should be emphasized that the attractiveness of meals offered in such school canteens was higher than that of the catering offers, in which the sensory quality of meals was lower while the prices were higher.

At the same time, a broad offer of unhealthy products, such as sweet and salty snacks and sugar-sweetened beverages, was still available in primary school shops and vending machines, which is inconsistent with the obligatory regulations.

To sum up, a recommendation can be made to develop a comprehensive food quality management program covering all points of offering foods at school, whose implementation would be monitored by designated people, e.g., school headmasters or their deputies, and state inspection services such as the Local Sanitary Inspectorate. Increasing the attractiveness of healthy foods and meals in school shops, vending machines, and canteens can contribute to healthier food choices and changes in the food preferences of children and adolescents.

## Figures and Tables

**Table 1 foods-13-00061-t001:** Canteen characteristics in schools participating in the research.

School Code	School Localization	Total No. of Pupils at School	Number of People Eating School Meals			Canteen	Canteen Workers	Number of Places in Canteen	Catering	School Shop	Vending Machine
			Pupils (Percentage of All)	Teachers	Including Free Meals						
1	A	222	200 (90.1)	5	0	X	6	100	-	-	-
2		955	320 (33.5)	6	5	X	6	50	-	X	-
3		420	205 (48.8)	3	12	X	4	70	-	-	X
4		785	440 (56.5)	4	15	X	4 *	78	-	X	-
5		650	380 (58.5)	0	5	-	4 *	66	X	-	X
6		630	560 (88.9)	20	24	X	5	84	-	-	X
7		910	520 (57.1)	0	86	X	6	96	-	X	X
8		560	380 (67.9)	0	4	X	3	50	-	-	-
9		700	180 (25.7)	0	1	X	3	76	-	-	-
10		300	170 (56.7)	0	5	X	3	100	-	-	-
11	B	1056	600 (56.8)	15	21	-	8	100	X	X	-
12		1009	500 (49.6)	10	27	X	6	60	-	X	X
13		415	150 (36.1)	12	70	X	5	50	-	-	-
14a		465	60 (12.9)	0	2	-	canteen (7) * trainees (2) *	20	X	-	-
14b		302	80–90 (29.8)	0	2	-		40	X	-	-
14c		830	200 (24.1)	0	3	-		50	X	X	-
14d		400	100–120 (25.0)	0	4	-		45	X	-	-
15		646	286 (44.3)	3	8	X	4	70	-	-	X
16		640	420 (65.6)	20	40	X	6	100	-	X	X
17a		350	20 (5.7)	0	5	-	5 (catering) *	50	X	X	-
17b		200	20–30 (15.0)	0	5	-		40	X	X	-
18	C	227	190 (70.4)	5	28	X	3	45	-	-	X
19		242	65 (26.9)	0	12	X	-	40	X	-	-
20		306	110 (9.1)	0	0	X	-	50	X	-	-
21	D	270	160 (59.3)	11	20	X	3	40	-	-	X
22		75	10 (13.3)	0	10	-	-	in common room	X	-	-
23		540	510 (94.4)	15	32	X	7	160	-	-	X
24		215	85 (39.5)	6	19	X	2	50	-	-	-

A—major cities with over 500,000 inhabitants; B—cities with populations ranging from 100,000 to 500,000; C—small towns with populations of 101,000 up to 50,000; D—villages with various populations ranging from 330 to 2000; * number of catering company workers for schools with catering; X: there is; -: there is not.

**Table 2 foods-13-00061-t002:** Meals offered in primary schools.

School Code	Working Hours	Meals Offered				Price of 1 Meal (PLN) **
		Breakfast	Lunch	Afternoon Snack	Special Diets	
1	7–15	-	X	-	gluten-free (1)	5
2	7–15	-	X	-	lack	8
3	6–14	-	X	-	lack	8
4	6–14	-	X	-	lack	10
5	6–14	X	X	X	lack	10 breakfast or afternoon snack: 3
6	6.30–14.30		X	X	lactose-free	6 with afternoon snack: 7
7	7–15	-	X	-	lack	8
8	7–15	X	X	X	lactose-free	lunch: 8 breakfast: 4 afternoon snack: 2 all 3 meals: 14
9	7–15		X		elimination diets	15
10	7–15		X		lack	15
11	7–15		X		lack	4.5
12	6–14		X		gluten-free (1)	5
13	6.30–14.30		X		lack	5
14a,b,c,d	7–15, including supplies 12–14 *	-	X	-	gluten-free (1)	10
15	7–15	-	X	-	lack	6
16	7–15	-	X	-	lack	5
17a,b	11–12 *	X	X	-	lack	lunch: 12 breakfast: 4.5
18	7–15	X	X	X	lack	4 all 3 meals: 9
19	7–15	-	X	-	lack	5
20	11–12 *	-	X	-	lack	10
21	6–14	-	X	-	lack	8
22	10–11 *	-	X	-	lack	17
23	7–16	X	X	X	lactose-free gluten-free vegetarian	7 all 3 meals: 12
24	7–15	X	X	X	3 people allergic to nuts	lunch: 7 breakfast: 2.5 afternoon snack: 2

* meals delivered and served by catering; ** soup, main course, beverage (water, compote, or other); X: there is; -: there is not.

**Table 3 foods-13-00061-t003:** Dish preferences among students.

Group of Dishes	Dish	Like/Dislike	Group of Dishes	Dish	Like/Dislike
Soup	varied, mostly vegetable	+	Meat dishes Main courses	pork cutlets (tenderloin, neck, pork chop)	++
	tomato, broth	++			
Bread	wheat–rye	+		gyros	+
	white, wheat	++		beef meat	− −
Starch Main course	potatoes	+		poultry in sauce	++
	rice	+		minced cutlets	++
	barley groats, bulgur	++		poultry cutlet/nuggets	++
	dumplings, pasta	++		fish	− −
	fries	++		liver	− −
Vegetable Main course	salads	− −	Meatless (sweet) dishes Main courses	sweet pasta/rice	++
	cooked	+		dumplings with cottage cheese with fruit	+ − ++
One-pot dish	spaghetti	++		potato–cottage cheese dumplings	++
	baked beans	+		pancakes, drop scones	++
	bigos	−		potato dumplings with fruit	− −
	casseroles	++	Beverages	fruit juices	+
	risotto	+		water	+
Dessert	fruit, yogurt	+		compote, tea	++
	cakes, donuts, bars, waffles	++		milk drinks with fruit	++

disliked (− −), eaten reluctantly (−), average liking depends on the child (+ −), liked (+), bestsellers (++).

## Data Availability

Data are contained within the article and Appendix A.

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
