# Peer review of "Do Food and Meal Organization Systems in Polish Primary Schools Reflect Students’ Preferences and Healthy and Sustainable Dietary Guidelines? The Results of Qualitative Research for the Junior-Edu-Żywienie (JEŻ) Project"

_foods, 2023, doi:10.3390/foods13010061_

Round 1

Reviewer 1 Report

Comments and Suggestions for Authors

Ewa Czarniecka-Skubina et al. submitted to Foods an article, focusing to a qualitative research performed in Poland on the students’ preferences and dietary guidelines.

The work described in the manuscript is well structured and is an expression of numerous organizational methods adopted in the country where the study was conducted, with a focus also on sustainability and food waste management.

24 schools out of more than 2000 participating in the project is not a representative number, and this could contribute to providing only partial data. This important limitation must be clearly indicated in the abstract and must be better specified in the limits of the study which is exclusively a pilot study, also to open up subsequent insights on an adequate sample size, and to provide more complete results coming from this important local project.

Comments on the Quality of English Language

Minor English editing required

Author Response

Dear Reviewer,

Thank you for reviewing our manuscript. A correction has been made in line with your  comments.

24 schools out of more than 2000 participating in the project is not a representative number, and this could contribute to providing only partial data. This important limitation must be clearly indicated in the abstract and must be better specified in the limits of the study which is exclusively a pilot study, also to open up subsequent insights on an adequate sample size, and to provide more complete results coming from this important local project.

 Thank you for your comments.

An in‐depth interview is a qualitative research technique that is used to conduct detailed interviews with a small number of participants. In case of in-depth interviews, it is common to conduct as few as 10 to 15 interviews [Rutledge, Hogg, 2020]. In our study we have 48 participants, including school headmasters (n=24), canteen workers (intendants, cooks) or representative of catering companies (n=24).

Such small numbers of participants are in case of the purpose of in‐depth interviewing is to get detailed information that sheds light on an individual's perspective, experiences, feelings, and the derived meaning about a particular topic or issue. Small sample size is one of the primary disadvantages/weaknesses of in-depth interviewing, but our research is completed by quantitative research, the results of which are not presented in this manuscript and will be published successively with reference to this article.

 As suggested by the reviewer, we have added the Limitation section:

Limitation

The conducted study has limitations that are inherent to qualitative research, which, due to its small sample sizes are not representative but help to better understanding quantitative outcomes. However, when combined with the results of quantitative research, as is the case in the "Junior-Edu-Żywienie (JEŻ)" project a deeper and more comprehensive explanation of the quantitative research findings was provided.

Rutledge, P.; Hogg, J.L. In-Depth Interviews. The International Encyclopedia of Media Psychology. Jan Van den Bulck(Ed.) John Wiley & Sons, Inc. 2020, 1-7. doi:10.1002/9781119011071.iemp0019

We hope that the improved manuscript will find your acceptance for publication. Thank you for your patience and help.

Authors

Reviewer 2 Report

Comments and Suggestions for Authors

This work is interesting. Some improvements should be considered to improve overall quality.

Pg. 5, lines 187-189 – “The percentage of children and teaching staff consuming meals at school most often depended on the quality of meals, including their organoleptic properties (palatability) and price.” Tables do not support this sentence. Please provide more data to support this sentence.

Pg. 7, line 270 – In section “3.4. How are the school meals planned?” authors should provide information about the number of nutritionists that are working in each school or in each catering company. They also should provide about information if all the meals in all schools are planned by nutritionists.

Pg. 12, lines 467-468 – In this sentence “Many schools introduced the system of canceling the school melas every morning due to the absence of the student what enabled to regulate.” Please replace melas by meals.

Pg. 12 lines 475-476 – “In rural areas, food waste was taken to feed animals, and it was estimated to amount approximately 10 Liters.” Please better clarify this sentence. 10 liters per day? In which school from the study?

Pg. 13, line 537 . “Canteen employees.” To support the discussion inside this section authors should provide according to literature the ideal ration of staff by students that is needed in canteens. And then, compare it with the ratios found in different schools within the study.

Discussion section: In the context of Poland, are there governmental guidelines outlining general rules for the structure and composition of school meals, as well as regulations for foodstuffs available in buffets and vending machines? The authors are encouraged to include this information in the discussion, drawing comparisons with guidelines from other countries to enrich the analysis.

Conclusion section: “the comprehensive food quality management program covering all points offering food at school” should be monitorized only by “school headmasters or their deputies”? Why don’t authors consider also to include food engineers and nutritionists?

Author Response

Dear Reviewer,

Thank you very much for the opportunity to submit the revised version of our manuscript (ID: foods-2771724, entitled: ‘Do food and meal organization systems in Polish primary school reflect students’ preferences and healthy and sustainable dietary guidelines? The results of qualitative research. The Junior-Edu-Żywienie (JEŻ) Project”).

We greatly appreciate the time and efforts taken by the Reviewers to review our manuscript. We have addressed all issues indicated in the review report, and believe that the revised version can meet the journal’s publication requirements.

All changes in the manuscript were marked using track changes option in Word or in red colour. We hope that the improved manuscript will find your acceptance for publication.

Thank you for your patience and help.

Kind regards

Authors       

Round 2

Reviewer 2 Report

Comments and Suggestions for Authors

Authors improved the manuscript by introducing corrections and clarifying the various aspects that were indicated by the reviewers.